# Planar Crack Approach to Evaluate the Flexural Strength of Fiber-Reinforced Concrete Sections

**DOI:** 10.3390/ma15175821

**Published:** 2022-08-24

**Authors:** Jacinto R. Carmona, Raúl Cortés-Buitrago, Juan Rey-Rey, Gonzalo Ruiz

**Affiliations:** 1Escuela Técnica Superior de Arquitectura de Madrid (ETSAM), Universidad Politécnica de Madrid (UPM), 28040 Madrid, Spain; 2Escuela Técnica Superior de Arquitectura de A Coruña, Universidad de A Coruña (UDC), 15071 A Coruña, Spain; 3E.T.S.I. Caminos, Canales y Puertos, Universidad de Castilla-La Mancha (UCLM), 13001 Ciudad Real, Spain

**Keywords:** cohesive fracture, fiber-reinforced concrete, size effect, brittleness number, FRC flexural strength

## Abstract

This article describes a model based on concepts of Fracture Mechanics to evaluate the flexural strength of fiber-reinforced concrete (FRC) sections. The model covers the need by structural engineers to have tools that allow, in a simple way, the designing of FRC sections and avoiding complex calculations through finite elements. It consists of an analytical method that models FRC post-cracking behavior with a cohesive linear softening law (*σ* − *w*). We use a compatibility equation based on the planar crack hypothesis, i.e., the assumption that the crack surfaces remain plane throughout the fracture process, which was recently proven true using digital image correlation. Non-cracked concrete bulk follows a stress–strain law (*σ* − *ε*) combined with the Bernoulli–Navier assumption. We define a brittleness number derived from non-dimensional analyses, which includes the beam size and the softening characteristics. We show that this parameter is key to determining the FRC flexural strength, characterizing fiber-reinforced concrete, and reproducing the size-effect of sections in flexure. Moreover, we propose an expression to calculate the flexural strength of FRC as a function of the cited brittleness number. The model also gives the ratio between the residual strength in service conditions and the flexural strength. Model results show a good agreement with tests in the scientific literature. Finally, we also analyze the brittle–ductile transition in FRC sections.

## 1. Introduction

Fiber-reinforced concrete (FRC) is a composite material characterized by a cementitious matrix and discrete fibers. The matrix is either concrete or mortar. Fibers can be of steel, polymers, carbon, glass, or natural materials and they provide post-cracking residual strength [1,2]. Thus, FRC can be considered inside the group of quasibrittle materials, characterized by the presence of a fracture process zone ahead of the real crack tip. The interest in the FRC as a structural material grew gradually throughout the last years, after the publication of design codes and recommendations in Europe [3], for example: [4,5,6,7]. Hence, there exists an increasing interest in designing tools for structural engineers regarding the application of FRC as a structural material [8].

The flexural strength in quasibrittle materials, such as FRC, does not coincide with the tensile strength and it is widely accepted that it is size-dependent [9]. Indeed, this size effect is included nowadays in some structural design concrete codes: Model Code 2010 [6,7]. The presence of a fracture process zone (FPZ) at the crack front, whose extension depends mainly on the microstructure of the material, has been accepted as the main cause of the transitional behavior from small to large sizes [10]. As FRC is a composite material characterized by a cement matrix and discrete fibers, the FPZ is larger than in plain concrete [11]. The main consequence is that the flexural strength in FRC is bigger than in plain concrete.

The post-cracking residual strength is defined by a softening function. Its implementation follows either (i) the fictitious crack model by [12], which uses a stress–crack width (*σ* − *w*) law to represent the softening, or (ii) the smeared crack band model firstly advanced by [13], which does so by a stress–strain width (*σ* − *ε*) relation. The latter depends on the width of the crack band [14]. In most cases, a computational FEM approach with tensile softening material behavior implemented is necessary to predict flexural strength in plain concrete and FRC [15,16,17].

Crack band models need to define a specific internal length (*L_i_*) to connect continuum mechanics, governed by a stress–strain constitutive relationship (*σ* − *ε*), and fracture mechanics, governed by a stress–crack opening relationship (*σ* − *w*). Hence, the aforementioned internal length is not a structural material property, but an artifact related to the length of the fracture process zone, the width of the band and the dimensions of the specimen. To properly perform this connection and prevent mesh dependency, several methods have related the internal length to physical parameters, such as maximum aggregate size for non-local approaches [17,18], or element size for local approaches [15,16]. In addition, several numerical crack models have been developed, such as the discontinuous numerical modeling of cracks using embedded discontinuities [19], the discrete strong discontinuity approach [20,21,22], dynamic fragmentation [23], the sequentially linear analysis method [24], or even a machine learning [25].

Based on this concept to relate discrete and continua media, a structural characteristic length (*l_cs_*) is defined as a parameter to convert stress–crack width (*σ* − *w*) curves into stress–strain (*σ* − *ε*) curves [26]. Although this structural length is used like the internal length (*L_i_*) mentioned above, it has a different meaning. While the first one, *L_i_*, is related to the distance between cracks and the depth of the neutral axis (macroscopic effects), the second one depends on the length of the fracture process zone (material properties) [27,28]. In all cases, these length parameters convey the idea that FRC is a continuous material, when in fact the main consequence of concrete post-cracking behavior is crack localization, which introduces a discontinuity in the material. Moreover, the *L_i_* parameter implies a structural dependence on the stress–strain curve (*σ* − *ε*), despite it should be totally independent of the type of structure considered.

To avoid the use of these length parameters, the present paper gives an analytical solution to evaluate flexural strength in FRC based on concepts of Fracture Mechanics. We model tension in cracked FRC by means of a softening law (*σ* − *w*). In this paper we use the linear function included in Model Code 2010 [3,7], but any other softening function included in codes or bibliography can be similarly adopted. The method uses a compatibility equation based on the planar crack hypothesis, e.g., on the assumption that the crack surfaces remain plane throughout the fracture process, which has been recently proven true by means of digital image correlation [29]. The compressive behavior of FRC is modeled through a linear elastic law (*σ* − *ε*) in conjunction with Navier’s hypothesis, applied only to the ligament. The crack opening is evaluated from the applied moment and the crack depth, obtaining a stress profile in the section in each crack step [30,31,32,33], using an expression proposed by [34]. Crack patterns are not evaluated because the model only considers a 1D section. The analytical solution fits tests results and expressions derived from cohesive models solved by finite element techniques.

One may wonder about the need for analytical solutions provided FEM-based expressions are already available. There are several reasons for this: (i) our analytical model is based on simple and well-known assumptions, (ii) in addition to the strength, it also gives the depth of the fracture zone and the stress profile distribution, (iii) its small- and large-size asymptotic behavior is correct, and (iv) it provides a better understanding of the relevant parameters [35]. So, the present paper aims at providing the theoretical frame for the planar crack approach in FRC sections and a tool to be used by structural engineers for designing FRC sections.

The paper is structured as follows. The subsequent section describes the material constitutive assumptions. Section 3 describes the crack propagation model. A detailed analysis of flexural behavior in FRC sections and its experimental validation is included in Section 4. Section 5 presents an analysis of the size effect in flexural strength. Section 6 shows a practical methodology based on model results to evaluate flexural strength and its validation range. Section 7 presents an analysis of the transition between ductile and brittle behavior. Finally, Section 8 summarizes the results of the paper and draws several conclusions.

## 2. Materials Hypothesis

The behavior of FRC is divided into two hypotheses depending on whether concrete tensile strength reached or not. It is considered one hypothesis for the non-cracked zone and another for the cracked zone.

### 2.1. Non-Cracked Zone

In the non-cracked area, concrete behavior is considered as an elastic material, which is represented by its elastic modulus. Navier’s hypothesis, e.g., planar sections remain planar after deformation, is used as a compatibility equation in the non-cracked area, see Figure 1. A linear stress distribution is adopted, where tension and compression stresses are proportional to the corresponding strain. A section without reinforcement bars and no compressive failure of concrete is assumed for this model. In the case of FRC sections without reinforced bars, during the crack progress, compression strength is only reached when the crack front is near to reach the beam depth [30]. Thus, compression failures normally occur after a long crack development.

### 2.2. Cracked Zone

Based on the cohesive model, a stress–crack opening law (*σ* − *w*) in uniaxial tension is defined as constitutive law for representing the post-cracking behavior of FRC. For the model development it is used the linear stress–crack opening law included in Model Code 2010 [7]; however, as aforementioned, the model can be adapted to any other softening function included in codes or bibliography. The *σ* − *w* used in the model presented in this paper is shown in Figure 2. The model in this paper is only valid for the case of post-cracking softening, meaning that a single discrete crack is localized in the FRC section. Post cracking hardening must be modeled using plasticity as no crack localizations will take place.

In Figure 2, *f_Fts_* represents the serviceability residual strength, defined as the post-cracking strength for serviceability crack openings, and *f_Ftu_* represents the ultimate residual strength. *f_Fts_* and *f_Ftu_* are calculated through the residual values of flexural strength by using the following equations:(1)$$f_{Fts}=0.45f_{R1}$$
(2)$$f_{Ftu}=f_{Fts}-\frac{w_{u}}{CMOD_{3}}\left(f_{Fts}-0.5f_{R3}+0.2f_{R1}\right)$$
where *f_R_*_1_ is the residual flexural strength corresponding to a crack mouth opening displacement (*CMOD*) of 0.5 mm and *f_R_*_3_ is the residual flexural strength corresponding to a *CMOD* of *w_u_*. These parameters are determined by performing a three-point bending test, on a notched beam, according to [36] (see Figure 3). *w_u_* is usually taken as 2.5 mm.

The ultimate tensile strength *f_Ftu_* in this linear model depends on the required ductility that is related to the allowed crack width. The ultimate crack width should not exceed 2.5 mm in any case. From Figure 2, the crack opening, *w*, in post-cracking constitutive law can be expressed as a function of the residual stress, *σ*:(3)$$w=\frac{f_{Fts}-\sigma }{f_{Fts}-f_{Ftu}}w_{u}$$

The area under the softening function is represented by *A_F,FRC_*. This area represents the theoretical energy required to open a unit area of crack surface, considering a linear softening.
(4)$$A_{F,\;FRC}=\frac{f_{Fts}+f_{Ftu}}{2}w_{u}\Rightarrow \;w_{u}=\frac{2A_{F,\;FRC}}{f_{Fts}+f_{Ftu}}$$

The planar crack assumption is used as a compatibility equation in the non-cracked area. This hypothesis was experimentally proven by [29]. The results of their study show that the crack propagation in the FRC predominantly occurs in the pre-peak and the post-peak softening response immediately following the limit of proportionality (LOP) as per [36]. Softening behavior in the load response immediately following the LOP is significantly influenced by the presence of fibers. The crack profile remains significantly planar after LOP, also when the crack depth approaches the beam height (formation of a localized hinge).

Based on the planar crack hypothesis, and in conjunction with a linear softening law, a linear stress profile can be considered during the crack processes on the cohesive ligament, see Figure 4.

## 3. Modeling of Crack Propagation

A rectangular FRC section is considered. The different geometric variables relevant to the problem are displayed in Figure 4. The section has a depth of *h*, and a width equal to *b*. The crack depth is represented as *z* and the neutral axis depth as *y_n_*.

All these dimensions can be expressed in a non-dimensional way by dividing them by *h*. In this manner, we define *ξ* = *z*/*h* as the non-dimensional crack depth, and *γ_n_* = *y*/*h* as the non-dimensional depth of the neutral axis; these parameters vary between 0 and 1. The non-dimensional crack opening is obtained by dividing it by the ultimate crack width, *w** = *w*/*w_u_*. 

The stress at the bottom of the section is named *σ_b_* and the stress at the top is *σ_t_*. Non-dimensional stresses are obtained by dividing by the serviceability residual strength, *f_Fts_*. So, we define *σ_b_** = *σ_b_*/*f_Ft_*_s_ and *σ_t_** = *σ_t_*/*f_Fts_*.

Crack propagation is divided into two different cases depending on the value of the crack opening at the mouth of the crack. FRC section is in case 1 when it is less than the ultimate crack width, *w_b_* < *w_u_*, and it is in case 2 when the crack mouth opening is bigger than the ultimate crack width *w_b_* > *w_u_*, see Figure 4.

In case 2, the crack depth for the critical opening is represented as *z_0_*. This value increases monotonically during the cracking process, so case 2 represents a decreasing curve [37] and for this reason, our study is focused on the development of case 1, where maximum load takes place.

The section equilibrium forces in case 1 can be expressed as:(5)$$\sum F=0\;\Rightarrow \frac{\sigma_{t}}{2}\left(h-y_{n}\right)b-\frac{f_{Fts}}{2}\left(y_{n}-z\right)b+\left(\frac{f_{Fts}+\sigma_{b}}{2}\right)zb=0$$

Expressing Equation (5) in a non-dimensional form, the following equation is obtained:(6)$$\sigma_{t}^{*}=\frac{\gamma_{n}+\sigma_{b}^{*}\xi }{1-\gamma_{n}}$$

The compatibility condition in the non-cracked zone is represented based on Navier’s hypothesis:(7)$$\frac{\varepsilon_{T}}{h-y_{n}}=\frac{\varepsilon_{ct}}{y_{n}-z}\Rightarrow \frac{\sigma_{t}}{h-y_{n}}=\frac{f_{Fts}}{y_{n}-z}$$

Expressing Equation (7) in a non-dimensional form allows deriving the following equation:(8)$$\gamma_{n}=\frac{1+\sigma_{t}^{*}\xi }{1+\sigma_{t}^{*}}$$

In the cracked zone, the constitutive law is formulated as:(9)$$w_{b}\left(M,z\right)=w_{b}\left(\sigma_{b}\right)$$

Crack opening, *w_b_* (*M*,*z*) can be evaluated by the expression given by [34]. *w_b_* (*σ_b_*) is defined considering the softening law, Equation (3). Thus, Equation (9) can be expressed as:(10)$$\frac{24M}{bh^{2}E_{c}}zf\left(\xi \right)=\frac{f_{Fts}-\sigma_{b}}{f_{Fts}-f_{Ftu}}w_{u}$$
where *f*(*ξ*) is the following shape function:(11)$$f\left(\xi \right)=0.76-2.28\xi +3.87\xi^{2}-2.04\xi^{3}+\frac{0.66}{{\left(1-\xi \right)}^{2}}$$

If we define a characteristic length as:(12)$$l_{ch,FRC}=\frac{E_{c}A_{F,FRC}}{f_{Fts}^{2}-f_{Ftu}^{2}}=\frac{E_{c}w_{u}}{2\left(f_{Fts}-f_{Ftu}\right)}$$

A brittleness number can also be defined as:(13)$$\beta_{H,FRC}=\frac{h}{l_{ch,FRC}}=\frac{2h\left(f_{Fts}-f_{Ftu}\right)}{E_{c}w_{u}}$$

This brittleness number has the same form of the Hillerborg’s brittleness number [8] but it is particularized for the case of linear softening with a residual stress, as it is shown in Figure 2. This brittleness number represents the size ratio between the section depth and the material characteristic length, which is a material property. Thus, Equation (10) in a non-dimensional form is expressed as:(14)$$\sigma_{b}^{*}=1-12M^{*}\beta_{H,FRC}\xi f\left(\xi \right)$$
where *M** is the bending moment in the section expressed in a non-dimensional form:(15)$$M^{*}=\frac{M}{bh^{2}f_{Fts}}$$

So, the bending moment in the section is equal to Equation (16):(16)$$M=\frac{1}{3}\sigma_{t}{\left(h-y_{n}\right)}^{2}b+\frac{1}{3}f_{Fts\;}{\left(y_{n}-z\right)}^{2}b+\left(\frac{f_{Fts}+\sigma_{b}}{2}\right)z\left[y_{n}-z\left(\frac{\frac{1}{3}f_{Fts\;}+\frac{1}{6}\sigma_{b}}{\frac{1}{2}\left(f_{Fts\;}+\sigma_{b}\right)}\right)\right]b$$

Equation (16) in a non-dimensional form is:(17)$$M^{*}=\frac{1}{3}\sigma_{t}^{*}{\left(1-\gamma_{n}\right)}^{2}+\frac{1}{3}{\left(\gamma_{n}-\xi \right)}^{2}+\left(\frac{1+\sigma_{b}^{*}}{2}\right)\xi \left[\gamma_{n}-\xi \left(\frac{2+\sigma_{b}^{*}}{3\left(1+\sigma_{b}^{*}\right)}\right)\right]$$

To evaluate the section stress profile, the crack opening, and the bending moment for a given crack depth, *ξ*, a system of four equations, Equations (6), (8), (14), and (17), can be solved analytically. The results of the equation system are *σ_b_**, *σ_t_**, *γ_n_*, and *M**, the only input data is *β_H,FRC_*. The crack depth, *ξ*, is used as a control parameter during the crack process. For each crack depth, only one equilibrium solution exists.

The crack opening at the bottom part of the section is evaluated as:(18)$$w_{b}^{*}=12M^{*}\beta_{H,FRC}\xi f\left(\xi \right)\left(\frac{1}{1-\alpha }\right)$$
where *α* is defined as the ratio between *f_Ftu_* and *f_Fts_*, (*f_Ftu_*/*f_Fts_*). Crack opening depends on the brittleness and on the *α* ratio previously defined. The maximum value for *w_b_** in case 1 is 1.0. Once this value is surpassed, case 2 applies.

## 4. Model Response and Experimental Validation

In this section, it will be shown how the brittleness number *β_H,FRC_* influences the behavior of the FRC section. In Figure 5a,b, the *x*-axis represents the non-dimensional crack mouth opening, *w_b_**, and the *y*-axis the non-dimensional bending moment during crack growth, *M**. In Figure 5a, the ratio between *f_Fts_* and *f_Ftu_*, *α*, has a constant value of 0.8. The initial point of all curves is the cracking moment. If we consider an elastic material, this crack has a non-dimensional value of 0.167. As the brittleness number decreases, peak load increases. Thus for smaller values of *β_H,FRC_* the softening length development is bigger. Therefore, this is the main reason for the increase in peak load. Section behavior is analyzed through moment versus crack opening curves, instead of the moment versus curvature curves, normally used in reinforced concrete section design. These curves give a more physical approximation to the FRC sections’ flexural behavior.

Figure 5b shows the influence of the ratio between *f_Ftu_* and *f_Fts_*, *α*, in the fiber reinforced concrete behavior. As in the previous case, the *x*-axis represents the non-dimensional crack mouth opening, *w_b_**, and the *y*-axis represents the non-dimensional bending moment during crack growth, *M**. The brittleness number *β_H,FRC_* has a constant value of 0.01 in the results shown. The peak load is not influenced by this parameter, as it is shown in the figure. As *α* increases, the nondimensional crack opening also increases. Therefore, when the slope of the softening curve decreases, so does the slope of the moment-opening curve after the peak.

In Figure 5a,b, it is observed that maximum load can be reached as an absolute maximum of the curve into the crack opening interval [0, 1], see, for example, the curve for *β_H,FRC_* = 0.01 and *α* = 0.4 in Figure 5b, or the maximum is reached in the interval limit, *w_b_** = 1.00, see, for example, the curve for *β_H,FRC_* = 0.1, and *α* = 0.8 in Figure 5a. This last case takes place when *w_u_** is reached at the bottom part of the crack, and the maximum is not within the interval [0, 1] of *w_b_**.

To validate the response of the model, we compared the results obtained with experimental results from the bibliography [38,39,40,41,42,43,44]. All of them correspond to contents of steel fibers of around 45–60 kg/m^3^, which is a usual range in fiber reinforced concrete elements. Figure 6 and Table 1 show the comparison, the *x*-axis represents the brittleness number, *β_H,FRC_*, and the *y*-axis the maximum non-dimensional bending moment during crack growth, *M_max_**. The model follows the experimental trends of experimental results with good agreement. Dotted horizontal lines represent the theoretical limits to the non-dimensional moment *M_max_** as will be explained in the next section.

This could be useful to take into account for FRC sections in the structural design, thus the prediction of the non-dimensional moment *M_max_** depending on the brittleness number, *β_H,FRC_*, is easy to obtain.

## 5. Size Effect on Flexural Strength for FRC

The flexural strength or modulus of rupture of a FRC section is defined as:(19)$$f_{R}=\frac{6M_{max}}{bh^{2}}=\frac{6M_{max}^{*}bh^{2}f_{Fts}}{bh^{2}}=6M_{max}^{*}f_{Fts}$$

In Figure 7a, the *x*-axis represents the brittleness number, *β_H,FRC_*, and the *y*-axis represents the non-dimensional flexural strength, *f_R_**, which is defined as the ratio between the flexural strength, *f_R_*, and *f_Ft_*_s_. The results obtained with the model are plotted with the expressions for the flexural strength evaluated by Uchida et al. [45] and Planas et al. [46]. These expressions were derived following a classical computational approach based on a cohesive constitutive law, in which secondary cracking is neglected [8]. Model results follow the same trend than the computational results, and they show the dependency of the flexural strength on the brittleness number, which represents the intrinsic size of the section.

Figure 7b shows the asymptotic behavior given by the model. The model response satisfies the asymptotic condition *f_R_** → 3 for *β_H,FRC_* → 0 (plastic limit solution for cohesive cracks, *M_max_** = 0.5) and *f_R_** → 1 for *β_H,FRC_* → ∞ (linear elastic solution, *M_max_** = 0.166). Moreover, the plot represents the usual brittleness number ranges for FRC and plain concrete. For FRC members, the flexural strength can be 150% to 200% higher than the tensile strength (*f_Fts_* in FRC) while for plain concrete this range is around 10–25%.

Note that for the lowest values of *β_H,FRC_*, the FEM approach shows an asymptotic behavior that does not satisfy the plastic limit solution, *f_R_** → 3, see Figure 7a. The existence of a non-negligible compressed area in the upper zone of the section stalls the crack growth, and therefore the plastic limit solution cannot be reached. Thus, for structural engineering purposes, it may be convenient to limit the value of *f_R_** to 2.5.

Size effect also can be understood through the non-dimensional stress profiles. In Figure 8, the *x*-axis represents the brittleness number, *β_H,FRC_*, and the *y*-axis represents the value of *σ_b_**, *γ_n_*, and *ξ* obtained with the proposed model.

We observe that, as *β_H,FRC_* increases, the crack depth and the depth of the neutral axis decrease. Crack depth shows an asymptotic trend of 0, and the neutral axis of 0.5, as is expected for the linear elastic solution. *σ_b_** also decreases as *β_H,FRC_* increases, but its influence on the flexural strength gets smaller because the softening zone also shrinks, compared to the depth, as *β_H,FRC_* increases. As an example of how the stress profiles vary, Figure 9 plots the non-dimensional stress distributions for several values of *β_H,FRC_*, namely 1, 0.1, 0.01, and 0.001.

The transition from the linear elastic to the plastic limit solution is seen in the non-dimensional stress profiles in Figure 9. For the usual brittleness number range of values for FRC (0.001–0.01), see Figure 7b, the crack depth has a value of over 0.6–0.7. So, we can conclude that the fracture process zone in FRC usually occupies most of the section height at maximum load. Thus, the scale effect in flexural strength in FRC is quantified and explained through the proposed model.

## 6. Practical Expression to Determine the Flexural Strength in FRC

Considering the equation form proposed by [46], the non-dimensional flexural strength, *f_R_**, can be expressed, fitting our model results, as:(20)$$f_{R}^{*}=1+\frac{1}{0.5+4.3\sqrt{\beta_{H,FRC}}}$$

It satisfies the asymptotic behavior discussed in Section 4. From Equations (19) and (20) is evaluated the ultimate non-dimensional bending moment, which is:(21)$$M_{max}^{*}=\frac{1}{6}+\frac{1}{3+25.8\sqrt{\beta_{H,FRC}}}$$

So, for structural purposes, the bending moment that a FRC section resists is expressed as:(22)$$M_{max}^{}=\left(\frac{1}{6}+\frac{1}{3+25.8\sqrt{\beta_{H,FRC}}}\right)bh^{2}f_{Fts}$$

All parameters included in this expression can be evaluated according to normalized tests or based on codes’ recommendations as Model Code 2010 [7] and new Eurocode 2 draft [47]. Safety factors also can be applied to FRC residual strength as described in the cited codes in order to use the design values *f_Fts,d_* and *f_Ftu,d_*.

Moreover, an expression to evaluate crack depth at maximum load (flexural strength) can be derived from the model predictions:(23)$$\xi_{max}=\frac{1}{1+5\sqrt{\beta_{H,FRC}}}$$

Figure 10a shows the results given by Equation (20) together with the results given by the model. Figure 10b shows the maximum non-dimensional crack depth versus the brittleness number.

Considering the residual strength *f_R1_* given in Table L.2: Residual Strength Classes for SRFC (Annex L) from Eurocode 2 draft [47] and Equation (22), it is possible to draw the maximum bending moment versus *f_R1_*. Figure 11 shows the variation of the curve depending on the ductility class. We considered two depths of the section (20 and 40 cm), for a characteristic compressive strength of concrete equal to 25 MPa. Elasticity modulus has been calculated according to Model Code 2010 [7].

The major influence resides in the depth of the section. There is an increment of roughly 300% between the bending moment for 20 cm and 40 cm of depth. On the other hand, the ductility class and the variation of the characteristic compressive strength are less relevant. So, these figures are practical to design the structural section according to the results from the tests.

The results given by Equations (20) and (23) are only valid when the maximum load is an absolute maximum in the non-dimensional crack interval [0, 1], as was explained in Section 4. So, based on the cited condition. Figure 12 shows graphically the value ranges of *β_H,FRC_*, and *α* where equations are valid.

Equations (20)–(23) cover ratios of *f_Ftu_*/*f_Fts_* between 0 and 0.8 for conventional FRC, which correspond with ratios *f_R_*_3_/*f_R_*_1_ in the range a–d according to Mode Code 2010 [7]. So, we conclude that the proposed expression covers most practical applications and can be used for structural design. In all cases, as mentioned in Section 5, for structural engineering purposes, it may be convenient to limit the value of *f_R_** to 2.5.

## 7. Brittle–Ductile Transition in Flexural Failure for FRC Sections

A minimum quantity of fiber in FRC elements can be determined to avoid brittle failure by imposing that the maximum cracking load, due to the matrix behavior, is lower than the ultimate load. This critical fiber quantity value is a limit that provides a ductile post-peak response of FRC members [48]. Figure 13 illustrates this brittle–ductile transition in a cracked FRC section.

The limit condition is:(24)$$M_{cr}=M_{max}$$
where *M_cr_* is the cracking moment of the concrete matrix and *M_max_* is described in Section 5 and can be evaluated from the model showed in this paper. Thus, Equation (24) can be expressed as:(25)$$f_{t}=f_{R}$$
where *f_t_* is the tensile strength of the concrete matrix and *f_R_* is the flexural strength or modulus of rupture. So, the behavior of the FRC section can be described in a non-dimensional form as:(26)$$f_{t}^{*}>f_{R}^{*}\;\mathrm{Brittle}\;\mathrm{behavior}$$
(27)$$f_{t}^{*}<f_{R}^{*}\;\mathrm{Ductile}\;\mathrm{behavior}$$
where *f_t_**, is equal to *f_t_*/*f_Fts_*. So, if *f_t_** is bigger than *f_R_**, fiber concrete section presents a brittle behavior, and, if *f_t_** is lower than *f_R_**, fiber concrete section presents a ductile behavior. Rearranging Equation (20) as a function of *β_H,FRC_*, and considering the limit condition in Equation (25), Equation (28) gives the value of the maximum brittleness number that has a ductile behavior for a given base concrete tensile strength. In other words, if a section has a brittleness number, *β_H,FRC_*, less than *β_H,FRC,max_*, for a given value of *f_t_**, this section will be ductile.
(28)$$\beta_{H,FRC,max}={\left[\frac{1}{4.3}\left(\frac{1}{f_{t}^{*}-1}-0.5\right)\right]}^{2}$$

Thus, the ductile or brittle response of a FRC section is determined by only two parameters, namely the tensile strength of the base concrete, *f_t_*, and the brittleness number, *β_H,FRC_*. In Figure 14, the *x*-axis represents the brittleness number, *β_H,FRC_*, and the *y*-axis the non-dimensional tensile strength of the base concrete, *f_t_**. Two areas are drawn defining the FRC section behavior (ductile or brittle). The boundary curve between brittle–ductile behavior is given by Equation (28).

## 8. Conclusions

An analytical model based on concrete Fracture Mechanics is presented to evaluate the flexural strength of FRC sections. The following conclusions can be drawn from the study:The planar crack assumption can be considered as an alternative to Navier’s hypothesis to model the FRC cracked zone. Using this approach, we avoid using length parameters as *l_cs_* to evaluate strains from crack openings, as is commonly carried out in models based on stress–strain laws.We propose a brittleness number, *β_H,FRC_*, analogous to the one of Hillerborg, as a characterization parameter of FRC structural sections. It is derived from a nondimensional analysis, which includes the beam size and FRC softening characteristics.The model fits experimental results very well. Moreover, the model reproduces the asymptotic behavior expected from plastic limit solution for cohesive cracks—very short depths—to the linear elastic solution—overly large depths.We offer an expression to calculate the flexural strength of a fiber-reinforced concrete section based on the model results. It depends on the brittleness number, *β_H,FRC_* and on the serviceability residual stress, *f_Fts_*. Its range of validity covers most of practical cases and thus, it can be profitably used for the structural design of FRC sections.The model, also, allows studying the ductile–brittle transition in FRC sections. It depends only on two parameters, namely the related tensile strength of the base concrete, *f_t_*, and the brittleness number, *β_H,FRC_*.The planar crack model contributes to a better understanding of the nature of flexural behavior of FRC sections and gives a more physical approach to their failure behavior. In addition, the expressions derived from the model results can be used for structural engineering purposes, constituting a design toolset that avoids complex calculations through finite elements.

## Figures and Tables

**Figure 1 materials-15-05821-f001:**
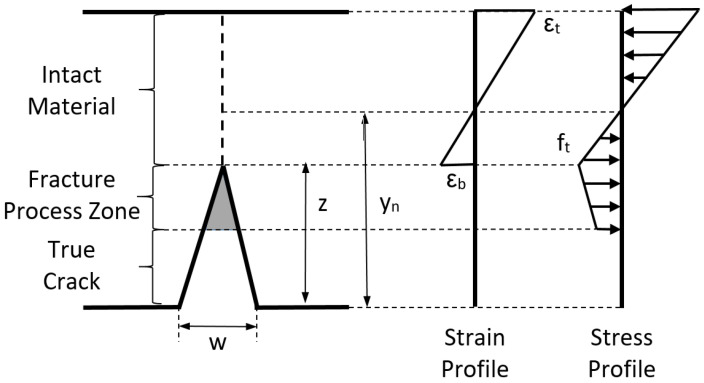
Material hypotheses. Cracked area is modeled according to the crack planar hypothesis and non-cracked area according to Navier’s hypothesis.

**Figure 2 materials-15-05821-f002:**
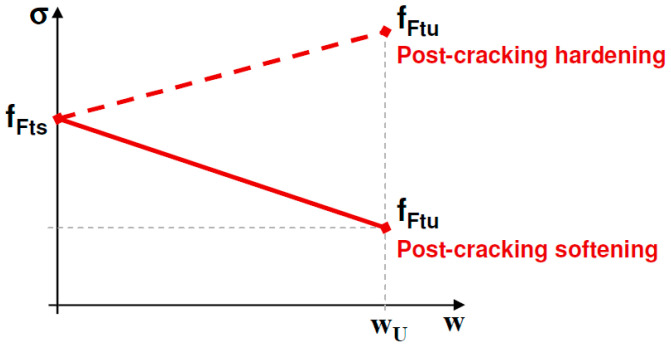
Simplified post-cracking constitutive law: stress–crack opening (softening post-cracking behavior). Model Code 2010 [7].

**Figure 3 materials-15-05821-f003:**
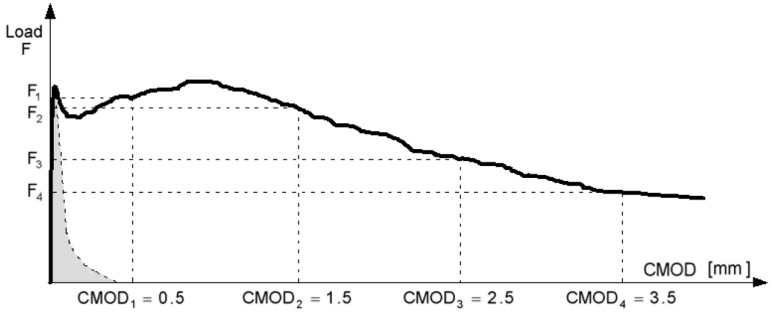
Applied load (*F*) versus crack mouth opening displacement (*CMOD*). Model Code 2010 [7].

**Figure 4 materials-15-05821-f004:**
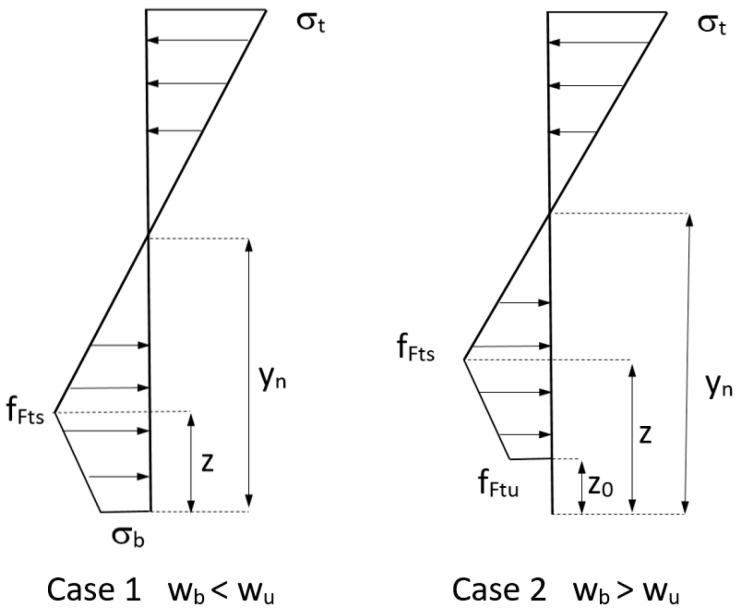
Crack propagation modeling cases.

**Figure 5 materials-15-05821-f005:**
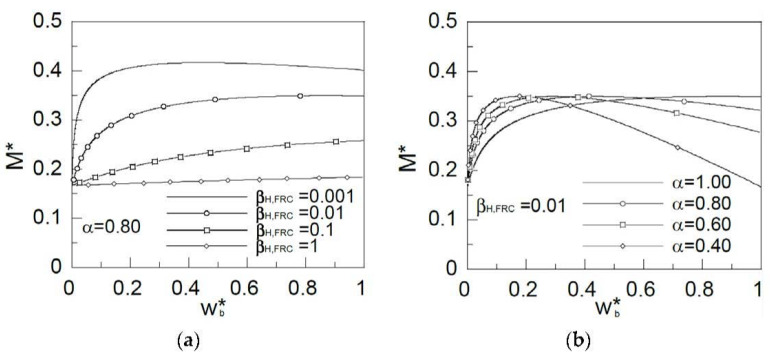
Influence of (**a**) the brittleness number, *β_H,FRC_*; and (**b**) the ratio between *f_Ftu_* and *f_Fts_*, *α*.

**Figure 6 materials-15-05821-f006:**
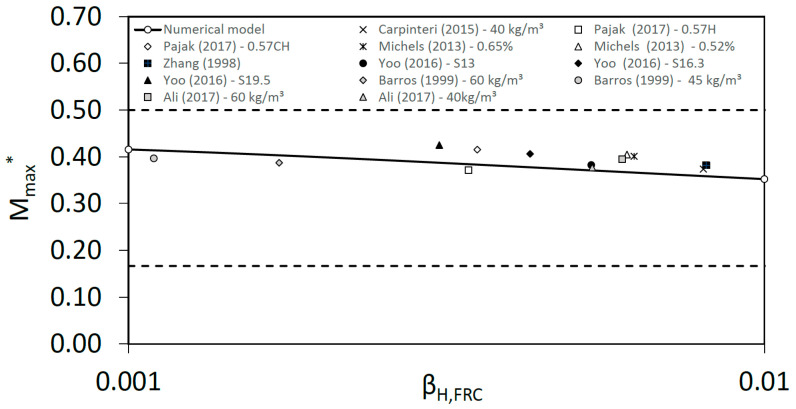
Non-dimensional maximum bending moment, *M_max_**, versus brittleness number, *β_H,FRC_*.

**Figure 7 materials-15-05821-f007:**
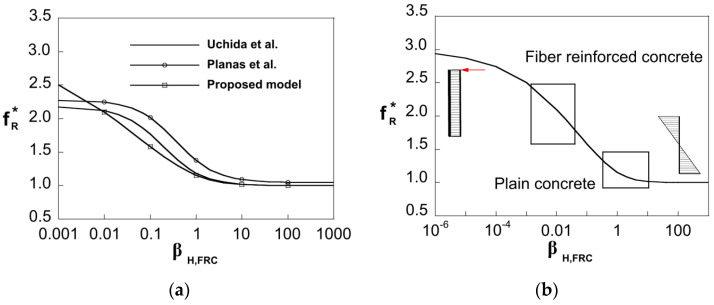
(**a**) Dependency of the flexural strength on the brittleness number, *β_H,FRC_*; (**b**) Asymptotic behavior.

**Figure 8 materials-15-05821-f008:**
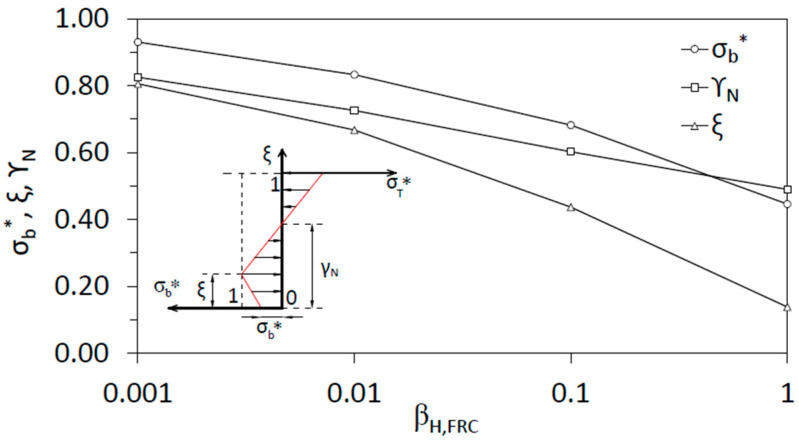
Non-dimensional stress at the bottom part of the beam, *σ_b_**, non-dimensional neutral axis depth, *γ_n_*, and non-dimensional crack depth, *ξ*, as a function of the brittleness number, *β_H,FRC_*.

**Figure 9 materials-15-05821-f009:**
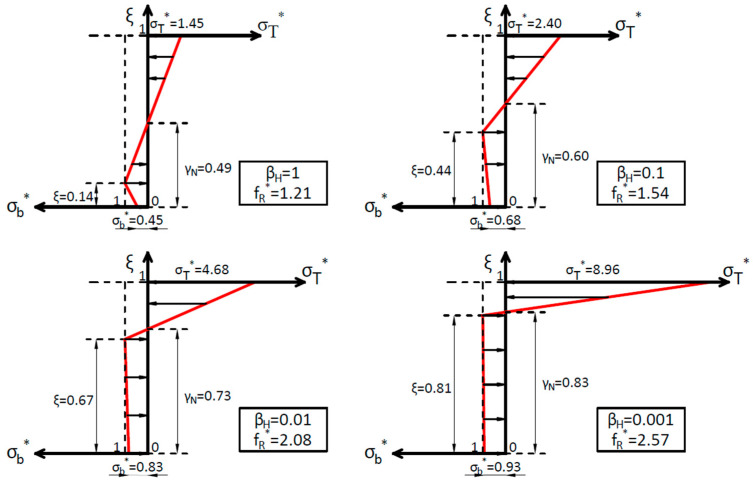
Non-dimensional stress profiles for *β_H,FRC_* = 1; 0.1; 0.01 and 0.001.

**Figure 10 materials-15-05821-f010:**
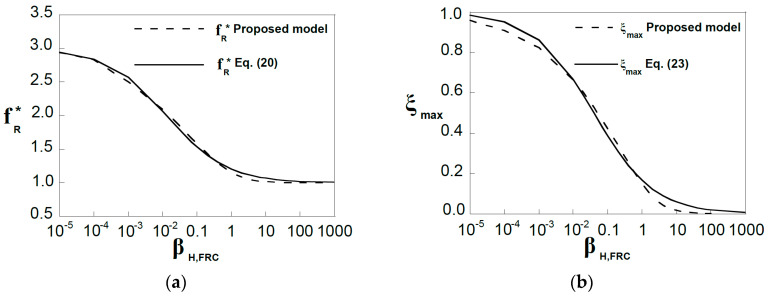
Planar crack model results for the non-dimensional flexural strength, *f_R_**, and corresponding crack-depth, *ξ_max_*, fitted by (**a**) *f_R_** given by Equation (20), and (**b**) *ξ_max_* given by Equation (22), respectively.

**Figure 11 materials-15-05821-f011:**
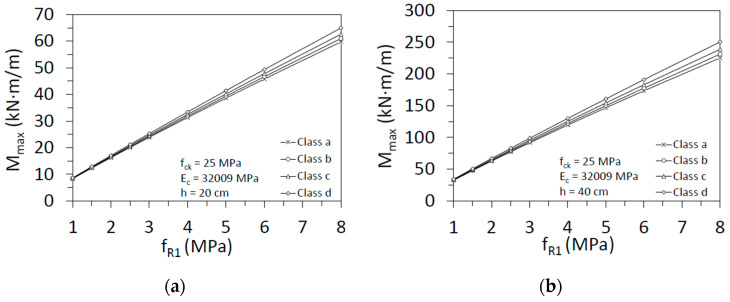
Maximum bending moment versus residual strength *f_R1_* for two section’s depths: (**a**) 20 cm; (**b**) 40 cm.

**Figure 12 materials-15-05821-f012:**
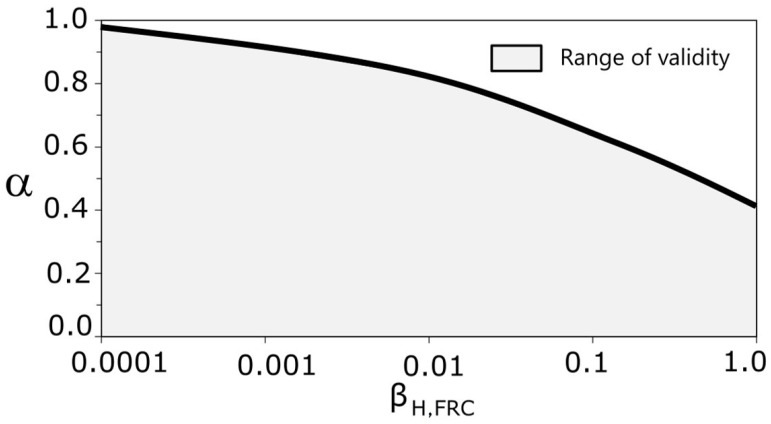
Validity range for the proposed formulation.

**Figure 13 materials-15-05821-f013:**
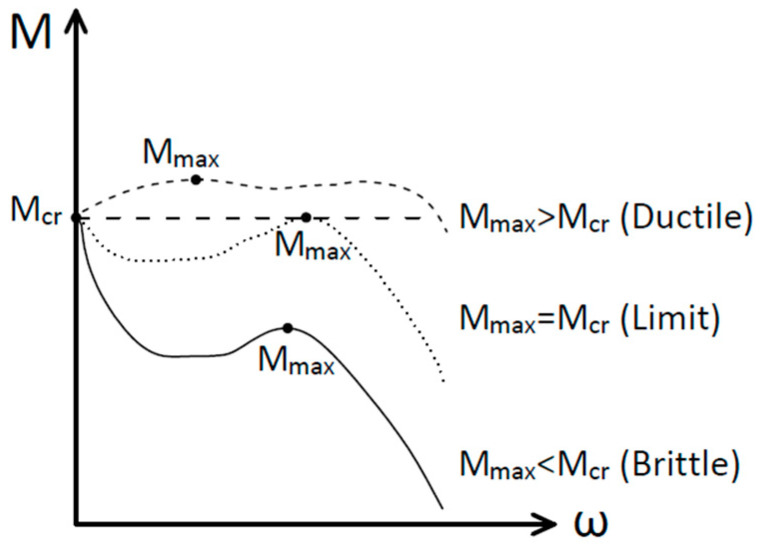
Types of behavior in an experimental test. Bending moment and crack opening.

**Figure 14 materials-15-05821-f014:**
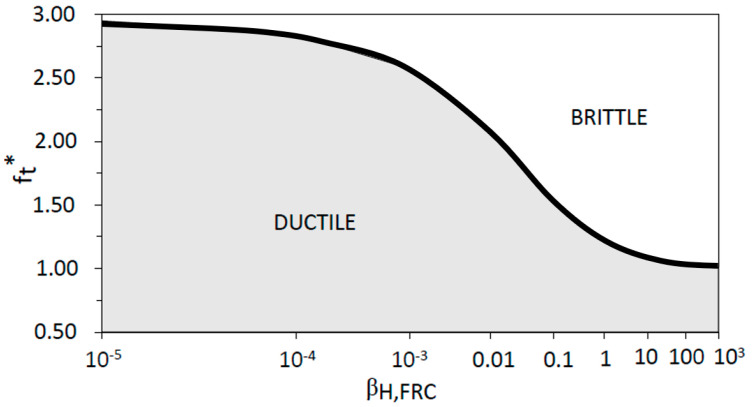
Transition between ductile and brittle behavior.

**Table 1 materials-15-05821-t001:** Experimental results from the bibliography compared in Figure 6 with numerical results from the planar crack model.

Reference	*V_f_* (kg/m^3^)	*P_max_* (kN)	*f_R,1_* (MPa)	*f_R,3_* (MPa)	*b* (m)	*h* (m)	L (m)	*E_c_* (MPa)	*w_u_* (mm)	*f_Fts_* (MPa)	*f_Ftu_* (MPa)	*A_f,FRC_* (N/mm)	*β_H_*
40 kg/m^3^ [38]	40	16.00	7.13	5.69	0.10	0.20	1.20	35,728	2.50	3.21	1.42	5.79	0.0080
Malgorzata-0.57H [39]	45	16.29	3.61	2.22	0.15	0.15	0.50	43,281	2.50	1.63	0.39	2.52	0.0034
Malgorzata-0.57CH [39]	45	16.49	3.27	1.78	0.15	0.15	0.50	41,903	2.50	1.47	0.24	2.13	0.0035
Michels. 0.65% [40]	51	31.40	7.73	6.93	0.15	0.15	0.60	30,000	2.50	3.48	1.92	6.75	0.0062
Michels. 0.52% [40]	41	30.60	7.47	6.67	0.15	0.15	0.60	30,000	2.50	3.36	1.84	6.50	0.0061
Zhang (78.4 kg/m^3^) [41]	78	16.50	9.60	6.00	0.10	0.10	0.40	32,000	2.50	4.32	1.08	6.75	0.0081
Doo-S13 (157 kg/m^3^) [42]	157	27.90	12.15	9.00	0.10	0.10	0.30	50,876	2.50	5.47	2.07	9.42	0.0053
Doo-S16.3 (157 kg/m^3^) [42]	157	32.90	13.50	12.60	0.10	0.10	0.30	46,260	2.50	6.08	3.60	12.09	0.0043
Doo-S19.5 (157 kg/m^3^) [42]	157	37.90	14.85	15.75	0.10	0.10	0.30	46,126	2.50	6.68	4.91	14.48	0.0031
Barros (60 kg/m^3^) [43]	60	11.50	2.20	1.90	0.15	0.15	0.45	33,366	2.50	0.99	0.51	1.88	0.0017
Barros (45 kg/m^3^) [43]	45	7.50	1.40	1.20	0.15	0.15	0.45	33,935	2.50	0.63	0.32	1.19	0.0011
Ali (60 kg/m^3^) [44]	60	28.00	7.00	5.67	0.15	0.15	0.60	34,484	2.50	3.15	1.43	5.73	0.0060
Ali (40 kg/m^3^) [44]	40	28.00	7.33	6.33	0.15	0.15	0.60	35,808	2.50	3.30	1.70	6.25	0.0054

## Data Availability

Not applicable.

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
