# Peer review of "Planar Crack Approach to Evaluate the Flexural Strength of Fiber-Reinforced Concrete Sections"

_materials, 2022, doi:10.3390/ma15175821_

Round 1
Reviewer 1 Report
The paper "Flexural strength and size effect of fiber reinforced concrete sections. Planar Crack Approach" is interesting, well written and adequate, but it needs some tweaking:
(i) The title has a point, it must be revised!!!
(i) The abstract must be reformulated to be more specific and present quantitative results;
(ii) The introduction can be improved with the addition of more recent works on the use of fibers in cementitious materials, for example: 10.1590/1807-1929/agriambi.v24n3p187-193; 10.1016/j.cscm.2022.e01297; 10.1016/j.cscm.2022.e01065.
(iii) "The response of ductile or brittle behavior comes given by only two parameters related to the plain concrete tensile strength of the matrix and the brittleness number, βH,FRC, which characterizes the behavior of the flexural strength of FRC section. In Figure 14, the x-axis represents the brittleness number, βH,FRC, and the y-axis represents the non-dimensional tensile strength of the concrete matrix, ft*. Two different areas are drawn defining the FRC section behavior. Transition brittle-ductile limit curve is also represented. The behavior of a section is determined by introducing in Figure 13, the value of ft* and βH,FRC" Explain this passage better.
(iv) "It is noted that the experimental results obtained correspond to quantities of fiber, in a range of steel fiber quantities around 45-60 kg/m3. This could be useful to take into account for FRC sections in the structural design. For lower quantities of fiber, the maximum load normally corresponding with the tensile strength of the concrete matrix, as will be analyzed in section 7" Explain this passage better!
(v) The conclusion must be reformulated by the authors.
Author Response
We are grateful to the reviewer for their detailed perusal and commentary of our work. Their comments have helped us to improve the paper. Below we give a detailed response to the questions raised in the reports. Answers are in italics below the reviewers' comments:
- The title has a point, it must be revised!!!
We revised the title and wrote it in one single sentence.
- The abstract must be reformulated to be more specific and present quantitative results;
The new version of the abstract is more specific and presents quantitative results.
- The introduction can be improved with the addition of more recent works on the use of fibers in cementitious materials, for example: 10.1590/1807-1929/agriambi.v24n3p187-193; 10.1016/j.cscm.2022.e01297; 10.1016/j.cscm.2022.e01065.
We included the suggested references along with additional ones to improve the paper.
- "The response of ductile or brittle behavior comes given by only two parameters related to the plain concrete tensile strength of the matrix and the brittleness number, βH,FRC, which characterizes the behavior of the flexural strength of FRC section. In Figure 14, the x-axis represents the brittleness number, βH,FRC, and the y-axis represents the non-dimensional tensile strength of the concrete matrix, ft*. Two different areas are drawn defining the FRC section behavior. Transition brittle-ductile limit curve is also represented. The behavior of a section is determined by introducing in Figure 13, the value of ft* and βH,FRC" Explain this passage better.
The new version of this passage explains Eq. (28) before Fig. 14. So, the reader can understand better the results plotted in this Figure (lines 372 to 377).
- "It is noted that the experimental results obtained correspond to quantities of fiber, in a range of steel fiber quantities around 45-60 kg/m3. This could be useful to take into account for FRC sections in the structural design. For lower quantities of fiber, the maximum load normally corresponding with the tensile strength of the concrete matrix, as will be analyzed in section 7" Explain this passage better!
We rewrote the passage to increase its readability (lines 245 to 252).
- The conclusion must be reformulated by the authors.
We reformulated the conclusions to facilitate the reader a better understanding of the subject and results.
Reviewer 2 Report
Overall, the topic of study is noteworthy and has filled the gaps of knowledge in the field of fibre-reinforced concrete. Before the paper can be accepted for publication, some minor improvements are required, and my comments are as follows:
1. Need to rework the title of the manuscript:
2. There was little information given on motivation, problem statements, validation methods and main findings in the abstract. I would reckon the abstract to be rewritten
3. The analytical method that signifies FRC post-cracking behaviour modelled with the linear softening law (σ-w) needs to be further expounded.
4. Why opted planar crack hypothesis
5. In the Introduction section, authors may give emphasis to different perspectives to validity in their review. This would provide an instructive presentation of review. Furthermore, the authors may also reorganize the reviews to highlight the impact of their Model response and experimental validation.
6. Standard code (BS or ASTM etc) should be explained.
7. For a more thorough debate on the proposed model, the authors are advised to discuss the possible reasoning for the observed trends with suitable facts or concepts prescribed in the relevant literature as well.
8. Regarding the addressed research area, it would be appropriate to also focus on the microstructure assessment (SEM analysis).
9. The conclusion in present form is a little scattered and resembles a summary of Section 3, 4 , 5 and 6 due to the absence of an authoritative research objective statement to guide the main conclusions.
10. References – consider the most recent publications in the subject area
Author Response
We are grateful to the reviewer for their detailed perusal and commentary of our work. Their comments have helped us to improve the paper. Below we give a detailed response to the questions raised in the reports. Answers are in italics below the reviewers' comments:
- Need to rework the title of the manuscript:
We rewrote the title to communicate better the content of the paper.
- There was little information given on motivation, problem statements, validation methods and main findings in the abstract. I would reckon the abstract to be rewritten
The new version of the abstract includes additional specific information about motivation, statements, validation, and results (lines 9-23).
- The analytical method that signifies FRC post-cracking behaviour modelled with the linear softening law (σ-w) needs to be further expounded.
We made some corrections in the introduction to clarify this point (lines 77 to 80; lines 124 to 128).
- Why opted planar crack hypothesis
The planar crack hypothesis is an alternative to Navier’s one as a compatibility condition to model the behavior of the FRC cracked zone. The main consequence of using this approach is to avoid the use of arbitrary length parameters as lcs, commonly used in models based on stress-strain constitutive laws. Besides, it allows using stress-opening laws that, in our opinion, give a more physical approximation to the FRC fracture behavior than stress-strain laws (Bažant, Z.P., and Planas, J. (1998). Fracture and size effect in concrete and other quasibrittle materials. Boca Raton: CRC Press.)
- In the Introduction section, authors may give emphasis to different perspectives to validity in their review. This would provide an instructive presentation of review. Furthermore, the authors may also reorganize the reviews to highlight the impact of their Model response and experimental validation.
We have rewritten the introduction to better justify the need for an analytical, physically based approach to study the failure of FRC sections. Likewise, we highlight the impact that the model may have in engineering practice.
- Standard code (BS or ASTM etc) should be explained.
The analytical model presented represents the tension in concrete through the linear softening law (σ-w) in Model Code 2010 [7] and [3]. However, the model can utilize any softening law in codes or bibliography. The new version of the paper includes this comment.
- For a more thorough debate on the proposed model, the authors are advised to discuss the possible reasoning for the observed trends with suitable facts or concepts prescribed in the relevant literature as well.
The paper includes a detailed size effect and asymptotic study and shows that the model results converge to these touchstone limit cases. We also compare our size effect with those obtained by Uchida et al. [45] and Planas et al. [46] through FE calculations. In addition, the paper validates the planar crack approach against several series of experimental results [39-44].
- Regarding the addressed research area, it would be appropriate to also focus on the microstructure assessment (SEM analysis).
The reviewer raises an interesting point since micro- and meso-resistant mechanisms of the material's internal structure influence the definition of the cohesive law, as pointed out in the introduction. The relation between these mechanisms and the constitutive laws constitutes a significant field of research. Many of the references in the paper deal with it, including some of the authors. However, here we focus on a sectional study that aims to be valid for practitioners and an alternative to the sectional calculation based on stress-strain laws used in codes.
- The conclusion in present form is a little scattered and resembles a summary of Section 3, 4, 5 and 6 due to the absence of an authoritative research objective statement to guide the main conclusions.
We rewrote the conclusions to communicate the paper subject and findings better. In addition, the new version of the paper highlights the principal research objective, namely providing a sectional model for FRC based directly on stress-crack opening laws.
- References – consider the most recent publications in the subject area
We included and commented on additional recent references to improve the paper.
Round 2
Reviewer 1 Report
ok.